

# A novel parameterized neutrosophic score function and its application in genetic algorithm

Yi Zhao[1], Fangwei Zhang[2], Bing Han[3], Jun Ye[4] and Jingyuan Li[2]

[1] College of Automotive and Traffic Engineering, Nanjing Forestry University, Nanjing, Jiangsu, China
[2] International Business School, Shandong Jiaotong University, Weihai, Shandong, China
[3] Warwick Manufacturing Group, University of Warwick, Coventry, United Kingdom
[4] School of Civil and Environmental Engineering, Ningbo University, Ningbo, China

## ABSTRACT

Efficiency, safety and cost are three major evaluation indexes of warehouse operation. However, the uncertainty of efficiency, safety and cost factors will lead to economic losses and waste of resources. The purpose of this study is to propose a novel parameterized neutrosophic objective–proportionate genetic algorithm model (PNO–PGA) to optimize the above three objectives. There are three main contributions of this study. Firstly, a novel score function of neutrosophic sets (NSs) is proposed to effectively integrate the fuzziness of efficiency, safety and cost to avoid the evaluation result being too idealized. Secondly, a novel proportionate genetic algorithm is applied to adaptively realize the iteration and inheritance processes. Finally, two parameters are proposed to make the algorithm model flexibly adapt to different types of environments and problems. Then, an example is used to compare the new method with genetic algorithm (GA). The result shows that PNO-PGA has better problem-solving ability in warehouse operation than GA.

## INTRODUCTION

With the rapid development of warehouse operations, the three objectives of efficiency, safety and cost constitute the core part of warehouse operation. Therefore, improving efficiency, reducing risks and increasing costs have received the attention of enterprises and researchers. Much exploratory research in the field of multi-objective have been done in the last few years. *Tanabe & Ishibuchi (2020)* established a set of 16 practical Multi-Objective optimization (MOO) problems with bounded constraints to make the performance evaluation more realistic. *Castonguay et al. (2023)* takes the modified MOO model to identify potential gains in efficiency in animal production. At present, the MOO method, which utilizes the robust optimization of the expectation and variance of the minimum function, is widely used in many fields, such as flood control of reservoirs (*Liu & Luo, 2019*), integrated energy system of buildings (*Wang et al., 2020*), corporate governance (*Jung & Choi, 2022*), risk-related resources scheduling (*Zuo, Zio & Xu, 2023*), *etc.*

Corresponding author
Fangwei Zhang,
fangweirzhang@163.com

Meanwhile, genetic algorithm (GA) has become an important method to solve MOO problem. *Costa-Carrapiço, Raslan & González (2019)* provided the first substantial evidence base for evaluating the potential of MOO by using GA. The performance of GA is improved by combining the advantages of different algorithms. In addition, GA can optimize the model, by using artificial neural network combing (*Yan et al., 2019*), sorting cuckoo search algorithm (*Aparna & Swarnalatha, 2023*) and simulation (*Perez-Tezoco et al., 2023*) together. The potential for improving MOO is realized by using GA technology. *Salata et al. (2020)* proposed a method for optimizing existing buildings structure by using GA technologies. Recently, *Dogan et al. (2022)* generalize a sludge biomass ash composting model based on deep neural network and genetic algorithm for sludge biomass ash composting optimization in the co-disposal process of dewatered sludge and biomass fly ash. Moreover, *Chen, Jia & He (2023)* put forward a novel bi-level multi-objective genetic algorithm to solve the integrated assembly line balancing and part feeding problem.

Many studies that apply simplified neutrosophic sets (NSs) to MOO show great research value. As a new fuzzy set, NSs use to improve the MOO model. Thereafter, *Rashno, Minaei-Bidgoli & Guo (2020)* proposed a clustering algorithm based on the NSs theory and a data uncertainty by exploring data density characteristics of the NSs. *Alpaslan (2022)* proposed a new NS-based complete local binary pattern hybrid method for texture classification. *Hassan, Darwish & Elkaffas (2022)* used Type-2 NSs to solve the medical database deadlock problem in real time. The existing study showed that, scholars have made significant progress in NSs, and how to make the objective function realistic has become a research hotspot. However, there are few researches on heuristic algorithms with multi-objective model which can adjust the weight flexibly by using NSs and expand the application ranges of the algorithm.

Simplified NSs have many advantages in handling uncertain information, and many scholars have conducted extensive research on this topic. *Ye (2014)* proposed a series of improved cosine similarity measures on simplified NSs. *Peng et al. (2014)* established a particular method for ordering to solve problems demand multi-criteria decision-making based on the outranking relation of simplified NSs. *Kilic & Yalcin (2020)* proposed a multiple stages methodology for sustainability performance evaluation by using NSs. *Huang et al. (2021)* aimed at the problem of ranking two single-valued neutrosophic sets (SvNSs) and proposed a method of ranking SvNSs values based on relative measures.

Moreover, the latest researches regarding to neutrosophic aggregation operators have also been reviewed. *Wu et al. (2018)* introduced a series of Hamacher aggregation operators on NSs. *Garg & Nancy (2019)* referred a new notion of possibility linguistic SvNSs. *Liu & Li (2019)* put forward generalized Maclaurin symmetric mean operators on NSs. *Yang, Fu & Han (2023)* developed a novel multi-criteria decision-making approach by using complying with the defined aggregation operators. *Kamran et al. (2023)* introduced an Einstein aggregation operator to handle uncertainties in the data. To advance the understanding of MOO problems, there are still certain issues, including reducing the computational cost of MOO problem and the effect of operators on target weights.

In a word, the extensive research and the aforementioned research results provide the theoretical basis for this study; moreover, scholars have made significant progress in

MOO problems and the solutions adopted. However, the research on warehouse operation management is not yet complete enough to structure a more complete framework to address the impacts and challenges of efficiency, safety, and cost on warehouse operation. Therefore, this study takes warehouse operation effectiveness as the object of research, and then, the efficiency, safety, and cost factors of production activities affecting enterprises are quantified. Meanwhile, it seems there is a contradictory and fuzzy relationship between efficiency, safety, and cost. When safety and efficiency are improved, costs may be significantly reduced as a result of optimized processes and reduced risk. However, improvements in safety may also have implications for both efficiency and cost. Moreover, changes in cost can directly drive alterations in efficiency and safety. Therefore, this study takes warehouse operation effectiveness as the object of research, and then, the efficiency, safety, and cost objectives of production activities affecting enterprises are quantified.

Specifically, this study combines the data processing ideas of NSs and heuristic algorithms together, and proposes a specific neutrosophic objective–proportionate genetic algorithm model (PNO–PGA). For convenience, the main research contributions are as follows: Firstly, this study is based on the concept of NSs, which integrates three different objectives of safety, efficiency, and cost affecting the production activities of enterprises to make the different objectives achieve compatible effects. Secondly, based on the NS, a generalized score function is introduced and applied to the proportional genetic algorithm to establish the PNO–PGA model, which provides a reference for evaluating and optimizing the production activities of enterprises. Finally, this study goes through a numerical example and calculates the model. Simulation software is used to compare the modified case with the unmodified case to get realistic results, and then verify the validity of the model.

The contents of this study are organized as follows: In "Preliminaries", the basic concepts of MOO, NSs, GA, and score functions of NSs are introduced. In "Optimization Method", the application of the generalized score function on SvNSs are described, and establish a parameterized neutrosophic goal-proportional genetic by using NSs. In "Illustrative Exemplicification", the effectiveness of the PNO-PGA model is validated using an example. In "Conclusion", the full text is summarized.

## PRELIMINARIES

**Definition 1**: Denote a universal set as $X$, for any given $x \in X$, denote

$$f(x) = \left(f_1(x), f_2(x), \ldots, f_k(x)\right), \tag{1}$$

$$e(x) = (e_1(x), e_2(x), \ldots, e_m(x)), \tag{2}$$

where $x = [x_1, x_2, \ldots, x_n]^T \in X$, $x_i^- \leq x_i \leq x_i^+$, wheras $x_i^-$ and $x_i^+$ are all crisp numbers. Then, a generalized MOO problem is described as following

$$\min f(x) = \left(f_1(x), f_2(x), \ldots, f_k(x)\right), \tag{3}$$

$$s.t. \, e(x) = (e_1(x), e_2(x), \ldots, e_m(x)) \geq 0. \tag{4}$$

Theoretically, it is difficult to find accurate feasible solutions for NP optimal problems in limited time. Then, the existing heuristics algorithms seek near-optimal solutions at a low cost, and GA is one of these heuristic algorithms. The specific definition of classical GA is as follows (*Lei et al., 2005*).

**Definition 2**: A GA is a kind of heuristic algorithm to solve optimization problems of both constrained and unconstrained. By referring the natural selection process in biology, the GA modifies one member of all individual solutions generation after generations. In each step, the new individuals for the next generation are generated by randomly selecting individuals from the current generation and using them as parents. Time and time again, the optimal solution can be obtained through population evolution. Through summary and induction, the aforementioned statement can be expressed as $GA = (C, E, P_0, M, \Phi, \Gamma, \Psi, N)$, where $C$ represents the coding methods of individuals, $E$ represents the evaluation function of the fitness value of individuals, $P_0$ represents the initial state of population, $M$ represents the size of population, $\Phi$ represents the operator for selection, $\Gamma$ represents the operator for crossover, $\Psi$ represents the operator for mutation, $N$ represents the end condition of GA. For convenience, a brief flow chart of GA is shown as Fig. 1.

In the following, the concept of classical single-valued NS is introduced. It is noteworthy that this kind of fuzzy set has been widely used in medical diagnosis (*Peng et al., 2014*), decision making (*Sodenkamp, Tavana & Caprio, 2018*), and image process (*Das et al., 2017*), etc.

**Definition 3**: Assume that $Y$ is a point space with a generic member represented by $y$. A truth membership function $T_A(y)$, an indeterminacy membership function $I_A(y)$, and a falsity membership function $F_A(y)$ form a single-valued neutrosophic set (SvNSs) $A$ on $Y$, where $T_A(y), I_A(y)$, and $F_A(y)$ are all mapping function from $Y$ to $[0, 1]$, whereas $0 \leq T_A(y) + I_A(y) + \Gamma_A(y) \leq 3$.

Then, a SvNSs $A$ is denoted as $A(y) = \{\langle x, T_A(y), I_A(y), F_A(y) \rangle | y \in Y \}$.

The single-valued neutrosophic element in $A$ is designated as $a = \langle T_A, I_A, F_A \rangle$. for convenience.

**Definition 4**: (*Guo, Şengür & Ye, 2014*) Assume that $Y$ contains two single-valued neutrosophic components, denoted as

$$a_1 = \langle T_A(y_1), I_A(y_1), F_A(y_1) \rangle, \tag{5}$$

$$a_2 = \langle T_A(y_2), I_A(y_2), F_A(y_2) \rangle. \tag{6}$$

Then, a classical similarity score between $a_1$ and $a_2$ is proposed as

$$S(a_1, a_2) = \frac{\left( T_S(y_1) \cdot T_S(y_2) + I_S(y_1) \cdot I_S(y_2) + F_S(y_1) \cdot F_S(y_2) \right)}{\left( \sqrt{T_S^2(y_1) + I_S^2(y_1) + F_S^2(y_1)} \cdot \sqrt{T_S^2(y_2) + I_S^2(y_2) + F_S^2(y_2)} \right)}. \tag{7}$$

In the following section, a novel PNO–PGA is proposed to solve MOO problem.

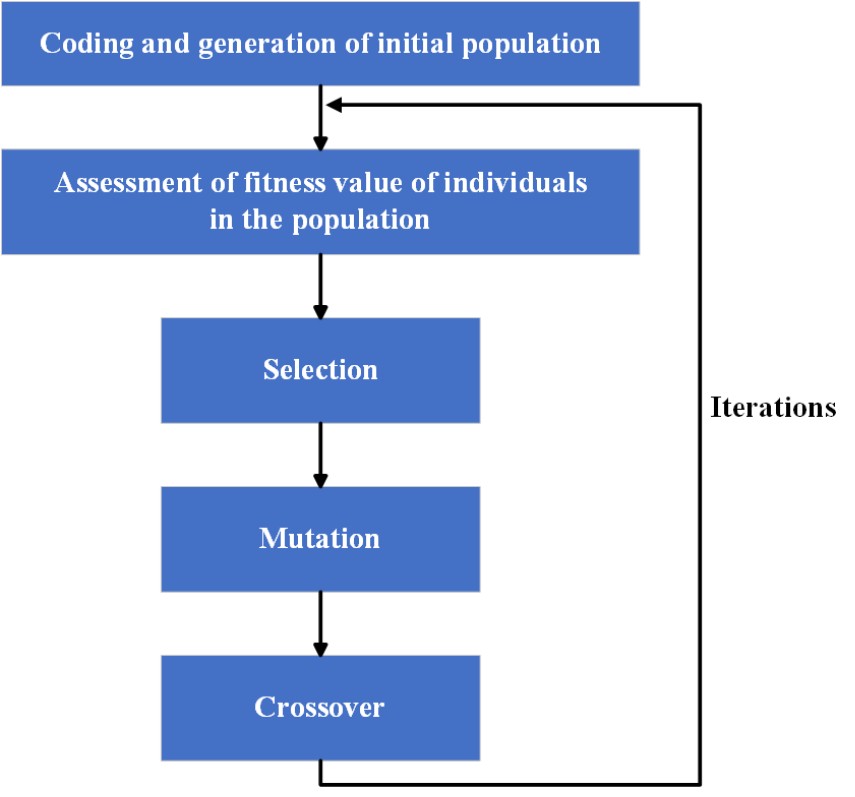

**Figure 1  Classical GA process.**

## OPTIMIZATION METHOD

### GSF on SvNSs

Inspired by *Guo, Şengür & Ye (2014)*, a kind of GSF on SvNSs is introduced as follows.

**Definition 5**: Assume that $a$ is a single-valued neutrosophic components on $Y$, it is denoted as

$$a = \langle T_A(y), I_A(y), F_A(y) \rangle. \tag{8}$$

Then, a kind of GSF on $a$ is denoted as

$$S(a) = f\left(T_A(y), \gamma\right), \tag{9}$$

where $f'_{T_A} > 0$, $f'_\gamma > 0$, $f\left(T_A(y), \gamma\right) \in [0, 1]$, whereas

$$\gamma = \cos\langle \left(T_A(y), I_A(y), F_A(Y)\right), (1, 0, 0) \rangle. \tag{10}$$

It is noteworthy that $S(\cdot)$ is composed of two independent variables, where one is $T_A(y)$ which indicates the modulus, whereas the other is $\gamma$ which indicates the degree of similarity between $a$ and $\langle 1, 0, 0 \rangle$.

**Definition 6**: Suppose $a$ is a single-valued neutrosophic number on $Y$, it can be denoted as

$$a = \langle T_A(y), I_A(y), F_A(y) \rangle. \tag{11}$$

Then, a specific GSF on $a$ is denoted as

$$S_F(a) = T_A(y) \cdot \gamma = T_A(y) \cdot \frac{T_A^\lambda(y)}{\sqrt{T_A^{2\lambda}(y) + I_A^{2\lambda}(y) + F_A^{2\lambda}(y)}}, \tag{12}$$

where $\lambda$ is a positive integer which represents the decision-maker's subjective attitude towards extremely large or small values. Obviously, $S'_{F,T_A} > 0$, $S'_{F,\gamma} > 0$, $S_F(\cdot, \cdot) \in [0, 1]$. Especially, when the weight vector for $\langle T, I, F \rangle$ is given as $\langle w_1, w_2, w_3 \rangle$, $S_F(a)$ can be expressed as

$$S_F(a) == \frac{T_A(y) \cdot (w_1 T_A(y))^\lambda}{\sqrt{(w_1 T_A(y))^{2\lambda} + (w_2 I_A(y))^{2\lambda} + (w_3 F_A(y))^{2\lambda}}}. \tag{13}$$

## PNO-PGA

On the basis of combining GA and NS theories, a novel heuristic algorithm is proposed. For convenience, the studied problem will be introduced firstly.

### (i) Problem introduction

In the production activities of enterprises, there are a variety of objectives to be considered, where efficiency, safety and cost are the three most important goals. For convenience, the measurement values of efficiency, safety and cost are denoted as $E$, $S$, and $C$, respectively. Assume that $E$, $S$, and $C$ are all bounded. Assume that there is a production project which has a collection of infinite alternatives. For convenience, the collection of alternatives is denoted as $X$, and the corresponding representative alternative is denoted as $x$. Denote $x = [x_1, x_2, \ldots, x_n]^T$, where $x_i^- \le x_i \le x_i^+$, $x_i^-$ and $x_i^+$ are all crisp numbers. For any given $x \in X$, it gets a vector of $E(x)$, $S(x)$, and $C(x)$. Here, as $E(x)$ and $S(x)$ get larger, the value $C(x)$ decreases, and the corresponding alternative $x$ is better. Denote the constraints in the given optimization problem as

$$e(x) = (e_1(x), e_2(x), \ldots, e_m(x)) \ge 0. \tag{14}$$

Denote

$$f^*(x) = (E(x), S(x), C(x)). \tag{15}$$

Then, a generalized MOO problem is described as

$$\min f^*(x) = (-E(x), -S(x), C(x)), \tag{16}$$

$$s.t. e(x) = (e_1(x), e_2(x), \ldots, e_m(x)) \ge 0. \tag{17}$$

### (ii) Problem analysis

To solve the aforementioned problem, the three objectives are explored before calculation. For any given alternative $x \in X$, it gets a vector $(E(x), S(x), C(x))$. Obviously, the larger $E(x)$ or $S(x)$, the better; the smaller $C(x)$, the better. Furthermore, $S(x)$ has an effect on $E(x)$ and $C(x)$. During production operations, when $S(x)$ is a small number, the management's attention is usually distracted by safety issues. To ensure the safety of production, the management will reduce the efficiency appropriately, which means the credibility of $E(x)$ will be reduced in real production. Meanwhile, when $S(x)$ is a small number, potential accident risks increases in the production process of industry, and these accident risks will be converted into the opportunity cost of production of enterprises, leading to the increase of $C(x)$. To eliminate the dimensional information from the data, $E(x), S(x)$, and $C(x)$ should be normalized. For convenience, this study denotes the normalized results as $E_N(x), S_N(x)$, and $C_N(x)$, respectively. Here, $E_N(x) \in [0, 1]$, $S_N(x) \in [0, 1]$, $C_N(x) \in [0, 1]$. Then, it can be obtained that $1 - S_N(x)$ is the potential risk of industry production. In this situation, the vector

$$G_{x_0} = \langle E_N(x), 1 - S_N(x), C_N(x) \rangle \tag{18}$$

is suitable to be dealt with as a simple neutrosophic number, and the set $X$ issuitable to be dealt with as a SvNSs, it gets

$$X_G = \{(x, \langle E_N(x), 1 - S_N(x), C_N(x) \rangle) | x \in X\}. \tag{19}$$

Here, for any given $x \in X$, by using Eq. (13), it gets its corresponding GSF value. By using the generalized score value, this study could optimize the given MOO problem.

### (iii) Parameterized PNO–PGA

Based on the problem analysis and by referring Definition 2, the introduced MOO problem (Eq. (17)) can be solved, and the detailed steps are shown as follows.

**Step 1**: For any given feasible solution $x$ for Eq. (17), by using $C$ in $SGA$, code $x$ and convert it into a vector $x'$ which is the representation of a feasible solution.

**Step 2**: Generate the initial population $P_0$ by using $x'$, and denote the population size of $P_0$ as $M$. Denote $X_0 = \{x_{01}, x_{02}, \dots, x_{0M}\}$, where $x_{0m} (1 \leq m \leq M)$ represents an individual. Denote $P_0 = \{x'_{01}, x'_{02}, \dots, x'_{0M}\}$, where $x'_{0m} (1 \leq m \leq M)$ is a vector which is corresponding to $x_{0m}$. The novel PNO–PGA will start the iteration with $P_0$. Set the iteration variable as $n$, and set the number of iterations as $N$.

**Step 3**: For any given $x_{0m} (1 \leq m \leq M)$ in $X_0$, its corresponding $E(x_{0m})$, $S(x_{0m})$, and $C(x_{0m})$ according to the actual condition in production practice are calculated. For the dangerous goods transport scenario mentioned in this article, $E(x_{0m})$, $S(x_{0m})$, and $C(x_{0m})$ according to Eqs. (39), (40), and (41) are calculated. Then, it gets the vector

$$V_{x_{0m}} = (E(x_{0m}), S(x_{0m}), C(x_{0m})). \tag{20}$$

For the whole initial population $P_0$, it gets a set of $G_{P_0}$. For convenience, the set is denoted as

$$V_{P_0} = \{V_{x_{01}}, V_{x_{02}}, \dots, V_{x_{0M}}\}. \tag{21}$$

Denote

$$\begin{cases} E_{MAX,0} = \{E(x_{01}), E(x_{02}), \ldots, E(x_{0M})\}, \\ S_{MAX,0} = \{S(x_{01}), S(x_{02}), \ldots, S(x_{0M})\}, \\ C_{MAX,0} = \{C(x_{01}), C(x_{02}), \ldots, C(x_{0M})\}. \end{cases} \tag{22}$$

For any given $m(1 \le m \le M)$. Denote

$$\begin{cases} E_N(x_{0m}) = E(x_{0m}) \cdot (E_{MAX,0})^{-1}, \\ S_N(x_{0m}) = S(x_{0m}) \cdot (S_{MAX,0})^{-1}, \\ C_N(x_{0m}) = C(x_{0m}) \cdot (C_{MAX,0})^{-1}. \end{cases} \tag{23}$$

Then, it gets a NS $G_0 = \langle E_N, 1 - S_N, C_N \rangle$. For any given $x_{0m}(1 \le m \le M)$, by using Eq. (23), it gets a neutrosophic number

$$G_{x_{0m}} = \langle E_N(x_{0m}), 1 - S_N(x_{0m}), C_N(x_{0m}) \rangle \tag{24}$$

and a set

$$X_G = \{(x, \langle E_N(x), 1 - S_N(x), C_N(x) \rangle) | x \in X\}. \tag{25}$$

**Step 4:** For any given $G_{x_{0m}}(1 \le m \le M)$, by using Eq. (13), its corresponding GSF value can be obtained as $S_F(G_{x_{0m}})$. Thereafter, by using $S_F(\cdot)$, a monotonic descending sequence according to $S_F(\cdot)$ is obtained as

$$l_{0F} = S_F\left(G_{x_{0m_1}}\right), S_F\left(G_{x_{0m_2}}\right), \ldots, S_F\left(G_{x_{0m_M}}\right), \tag{26}$$

where $m_1, m_2, \ldots, m_M$ isa permutation of $1, 2, \ldots, M$.

Set a positive integer $M^*(M^* < M)$. By using $M^*$, $X_0$ is divided into two parts, where one is

$$X_{01} = \left\{x_{0m_1}, x_{0m_2}, \ldots, x_{0m_{M^*}}\right\}, \tag{27}$$

and the other is

$$X_{02} = \left\{x_{0m_{M^*+1}}, x_{0m_{M^*+2}}, \ldots, x_{0m_M}\right\}. \tag{28}$$

**Step 5:** By using certain crossover operator $\Gamma$, $X_{01}$ is transferred to $X_{01}'$, where $X_{01}' = \left\{x_{0m_1}', x_{0m_2}', \ldots, x_{0m_{M^*}}'\right\}$. By using mutation operator $\Psi$, $X_{02}$ is transferred to $X_{02}'$, where $X_{02}' = \left\{x_{0m_{M^*+1}}', x_{0m_{M^*+2}}', \ldots, x_{0m_M}'\right\}$. Denote $X_0' = \left\{X_{01}', X_{02}'\right\}$ by using $S_F(\cdot)$, a monotonic descending sequence is obtained as

$$l_{0F}' = S_F\left(G_{x_{0m_1}'}\right), S_F\left(G_{x_{0m_2}'}\right), \ldots, S_F\left(G_{x_{0m_M}'}\right). \tag{29}$$

For any given $m(1 \le m \le M)$, denote

$$S_F(G_{x_{1k}}) = \max\left\{S_F\left(G_{x_{0m_k}}\right), S_F\left(G_{x_{0m_k}'}\right)\right\}, \tag{30}$$

denote the corresponding set of feasible solution $X_1$, where $X_1 = \{x_{11}, x_{12}, \ldots, x_{1M}\}$, and denote the *1* st optimal solution as $x_{11}$.

**Step 6**: If $N \neq 1$, then, turn to Step 3; and denote $n = 2$, denote

$$\begin{cases} E'_{MAX,1} = \max\{E_{MAX,0}, E_{MAX,1}\}, \\ S'_{MAX,1} = \max\{S_{MAX,0}, S_{MAX,1}\}, \\ C'_{MAX,1} = \max\{C_{MAX,0}, C_{MAX,1}\}. \end{cases} \tag{31}$$

Denote

$$\begin{cases} E_N(x_{1,m}) = E(x_{1,m}) \cdot (E'_{MAX,1})^{-1}, \\ S_N(x_{1,m}) = S(x_{1,m}) \cdot (S'_{MAX,1})^{-1}, \\ C_N(x_{1,m}) = C(x_{1,m}) \cdot (C'_{MAX,1})^{-1}. \end{cases} \tag{32}$$

Then, turn to Step 4, Step 5, Step 6 in turn. If $n < N$, turn to Step 3; and denote $n = n+1$,

$$\begin{cases} E'_{MAX,n} = \max\{E'_{MAX,n-1}, E_{MAX,n}\}, \\ S'_{MAX,n} = \max\{S'_{MAX,n-1}, S_{MAX,n}\}, \\ C'_{MAX,n} = \max\{C'_{MAX,n-1}, E_{MAX,n}\}. \end{cases} \tag{33}$$

Denote

$$\begin{cases} E_N(x_{n,m}) = E(x_{n,m}) \cdot (E'_{MAX,n})^{-1}, \\ S_N(x_{n,m}) = S(x_{n,m}) \cdot (S'_{MAX,n})^{-1}, \\ C_N(x_{n,m}) = C(x_{n,m}) \cdot (C'_{MAX,n})^{-1}. \end{cases} \tag{34}$$

Then, turn to Step 4, Step 5, Step 6 in turn. If $t < N$, turn to Step 3; if else, select the final optimal solution as $x_{N1}$. It is noteworthy that there is a series of virtual NSs which is structured by Eq. (22), and is denoted as $V_n = \{\langle E_n, 1-S_n, C_n\rangle \,|\, n = 1, 2, \ldots, N\}$. For any given individual $x_{n,m}$, the neutrosophic number corresponding to it is denoted as $V_{n,m} = \langle E_N(x_{n,m}), 1-S_N(x_{n,m}), C_N(x_{n,m})\rangle$. For ease of understanding, please refer to Fig. 2.

### (iv) Supplement explanations

(1) In Step 3, by using Eqs. (22)–(23) and (34), a series of NSs are structured. The characteristic of the proposed NSs is that they converge to a fuzzy set. Details on this convergence are introduced in the following theorem.

**Theorem 1**: For any given $n \leq N$, $m \leq M$, $M = +\infty$, it gets a neutrosophic number

$$V_{n,m} = \langle E_N(x_{n,m}), 1-S_N(x_{n,m}), C_N(x_{n,m})\rangle. \tag{35}$$

Denote

$$\lim_{n \to +\infty} V_{n,m} = \left\langle \lim_{n \to +\infty} E_N(x_{n,m}), 1 - \lim_{n \to +\infty} S_N(x_{n,m}), \lim_{n \to +\infty} C_N(x_{n,m}) \right\rangle. \tag{36}$$

Then, it gets that $\lim_{n \to +\infty} V_{n,m}$, $\lim_{n \to +\infty} E_N(x_{n,m})$, $\lim_{n \to +\infty} S_N(x_{n,m})$, and $\lim_{n \to +\infty} C_N(x_{n,m})$ all exist.

**Proof**: By using Eq. (31), it gets $E'_{MAX,1}$, $S'_{MAX,1}$, $C'_{MAX,1}$ all increase monotonically as $n$ increases, while $E(x_{1,m})$, $S(x_{1,m})$ and $C_N(x_{1,m})$ are positive crisp values. Then, by using monotone bounded theorem, it gets $\lim_{n \to +\infty} E_N(x_{n,m})$, $\lim_{n \to +\infty} S_N(x_{n,m})$, and

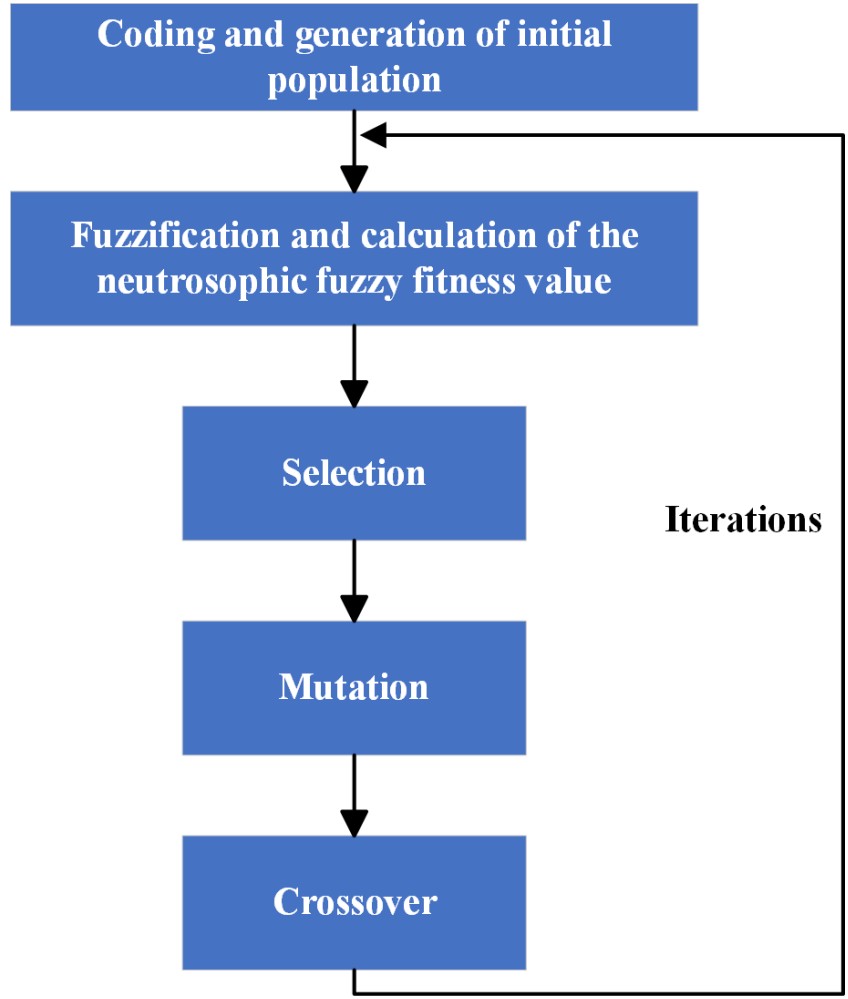

**Figure 2  Novel PNO-PGA process.**

$\lim_{n\to+\infty} C_N\left(x_{n,m}\right)$ all exist. Moreover, by using Eq. (36), it gets that $\lim_{n\to+\infty} V_{n,m}$ exists. (2) In Step 3, the neutrosophic fitness value is calculated by using Eq. (13). Like other classical score functions, Eq. (13) has its scope of application and defects. In specific optimization environment, users need to choose the appropriate score function according to their own practice. (3) In Step 4, the neutrosophic fitness value for each feasible solution is calculated by using Eq. (2). In this step, the parameter $\lambda$ can reflect the subjective attitude of the customer towards the relationship between efficiency, safety and cost. (4) In Step 4, for any given $G_{x_{0m}}$ $(1 \le m \le M)$, $S_F(\cdot)$ iused to calculate the neutrosophic fitness function value. Since $S_F(\cdot)$ is obtained by using fuzzy technology, for the same feasible solution, it can get different neutrosophic fitness function values in different iteration times. The reason for this phenomenon is that $E_{MAX}, S_{MAX}$ and $C_{MAX}$ are all variables in Eq. (22). (5) In Step 5, by suing $S_F(\cdot)$ and $M^*$, the individuals are divided into two parts, where one part is to be crossover processed, and the other is to be mutated. Here, it is noteworthy to point out that crossover and mutation are probabilistic in classical GA, while crossover and

**Table 1  Results comparison.**

| Algorithms | PNO-PGA | Basic GA |
|---|---|---|
| Maximum convergence iterations | 461 | 280 |
| Mean convergence iterations | 389 | 237.55 |
| Standard deviation of convergence iterations | 40.37 | 41.66 |
| Maximum CPU time(s) | 415.17 | 469.47 |
| Mean CPU time(s) | 405.59 | 416.23 |
| Standard deviation of CPU time (s) | 6.39 | 25.05 |

mutation are proportional in PNO–PGA. Especially, the crossover rate is $M^*/M$, while the mutation rate is $1 - M^*/M$. Moreover, the selected crossover individuals all take part in the crossover process, while the selected mutation individuals all take part in mutation process, which makes full use of the computing power of modern computer and reduce the occurrence opportunity of premature.

The differences between the proposed PNO–PGA and the GA are listed in Table 1.

## ILLUSTRATIVE EXEMPLIFICATION

### Exemplification introduction

To illustrate a MOO problem in warehouse operation, we cited the data which were collected as previously described in *Zhang et al. (2023)*. Assume that the length, width, and entrance width of the warehouse are 64 m, 17.5 m, and 4 m respectively. And there are 14 points inbound and points outbound. Not only should a distance of 12 m between two container vehicles but also a distance from the warehouse to the container truck be 2 m. For convenience, the inbound point is denoted as $\Delta_i (1 \le i \le 14)$, the outbound point is denoted as $\nabla_j (1 \le j \le 14)$. In general, the work flow of the forklift is as follows. Firstly, the hazardous goods are transported from containers by the operating forklifts. Secondly, the goods are unloaded by forklifts to the inbound point. Thirdly, the outbound goods should be discharged onto the outbound container by an empty forklift traveling to the outbound port. Assuming that there are three forklifts responsible for both inbound and outbound hazardous materials transfer. The forklift turns back to the inbound container and prepares the next inbound activity. For convenience, the chain of the above operations is named as a closed-loop storage chain. The forklifts work on successive chains until operations are over. According to *Sun et al. (2021)*, the original warehouse layout structure diagram is shown in Fig. 3.

The distance for forklifts to travel to finish a closed-loop storage chain depends on the location of the point incoming and the point outbound. For any given $\Delta_i$ and $\nabla_j$, there is a corresponding closed-loop storage chain whose distance is denoted as $l_{ij}$. Denote the distance of all pairs of $(\Delta_i, \nabla_j)(1 \le i, j \le 14)$ as $L = (l_{ij})_{14 \times 14}$ where the incoming point is represented by the row in L, while the outgoing point is represented by the column in L. To simplify the calculation, the original distances of the closed-loop storage chains are

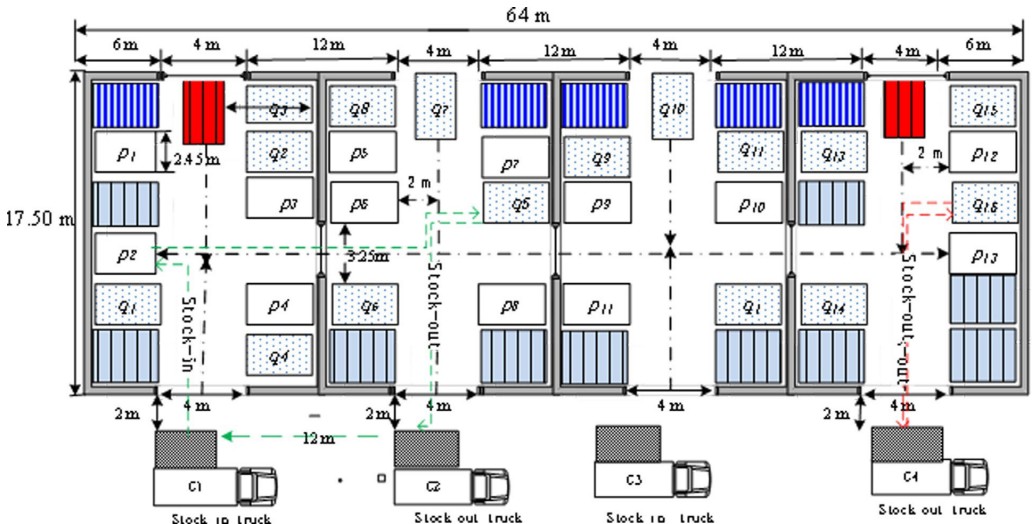

**Figure 3** The original warehouse layout structure diagram.

normalized by using the function

$$l'_{p_{i_{k1}},q_{j_{k2}}} = \frac{l_{p_{i_{k1}},q_{j_{k2}}} - \min_{S1,S2 \in M} l_{p_{i_{S1}},q_{j_{S2}}}}{\max_{S1,S2 \in M} l_{p_{i_{S1}},q_{j_{S2}}} - \min_{S1,S2 \in M} l_{p_{i_{S1}},q_{j_{S2}}}} \vee 0.1. \tag{37}$$

For any given $l'_{p_{i_1},q_{j_1}}, l'_{p_{i_2},q_{j_2}}, l'_{p_{i_3},q_{j_3}}$, the similarity degree $r'$ on them can be obtained as

$$r' = 1 - \frac{\left|\left(l'p_{i_1},q_{j_1} - l'p_{i_2},q_{j_2}\right) \cdot \left(l'p_{i_1},q_{j_1} - l'p_{i_3},q_{j_3}\right) \cdot \left(l'p_{i_2},q_{j_2} - l'p_{i_3},q_{j_3}\right)\right|}{\left(\max\left\{l'p_{i_1},q_{j_1} \cdot l'p_{i_2},q_{j_2} \cdot l'p_{i_3},q_{j_3}\right\}\right)^3}. \tag{38}$$

Denote the fixed cost for operation management per hour as $C_r$, denote the normal speed index of the forklifts as $\tau_v$, denote the time cost of the operation as $C_t = C_r \cdot \tau_v^{-1} \cdot l'$, denote the influence factor of variable speed on operation time as $\tau_\alpha$, denote the influence factor of variable speed on fuel cost as $\tau_\beta$, anddenote the normal fuel cost index for forklift to travel as $C_f$. Then, according to field survey, the efficiency optimization model can be expressed as

$$\begin{cases} \gamma_1 = (\tau_v)^{-1} \sum_{i_s=1}^{14} \sum_{j_t=1}^{14} x_{i_sj_t} \cdot l'_{i_sj_t} - \tau_\alpha \cdot \left(C_{14}^3\right)^{-1} \cdot \sum_{i_s,j_t=1}^{14} \left(x_{i_s,j_t,1} \cdot x_{i_s,j_t,2} \cdot x_{i_s,j_t,3} \cdot r'_{\left(p_{i_1},q_{j_1}\right),\left(p_{i_2},q_{j_2}\right),\left(p_{i_3},q_{j_3}\right)}\right), \\ s.t. \sum_{i_s=1}^{14} x_{i_sj_t} = 1, \sum_{j_t=1}^{14} x_{i_sj_t} = 1, \sum_{i_s=1}^{14} x_{i_s,j_t,1} = 1, \sum_{j_t=1}^{14} x_{i_s,j_t,1} = 1, \\ \sum_{i_s=1}^{14} x_{i_s,j_t,2} = 1, \sum_{j_t=1}^{14} x_{i_sj_t,2} = 1, \sum_{i_s=1}^{14} x_{i_s,j_t,3} = 1, \sum_{j_t=1}^{14} x_{i_sj_t,3} = 1, \\ x_{i_s,j_t,1} \neq x_{i_s,j_t,2} \neq x_{i_s,j_t,3}, x_{i_sj_t} \in \{0,1\}, x_{i_s,j_t,1} \in \{0,1\}, x_{i_s,j_t,2} \in \{0,1\}, x_{i_s,j_t,3} \in \{0,1\}. \end{cases} \tag{39}$$

The safety optimization model can be expressed as

$$
\begin{cases}
\gamma_2 = \left(C_{14}^3\right)^{-1} \cdot \displaystyle\sum_{i_s, j_t = 1}^{14} \left( x_{i_s, j_t, 1} \cdot x_{i_s, j_t, 2} \cdot x_{i_s, j_t, 3} \cdot r'_{\left(p_{i_1}, q_{j_1}\right), \left(p_{i_2}, q_{j_2}\right), \left(p_{i_3}, q_{j_3}\right)} \right), \\[2mm]
s.t. \displaystyle\sum_{i_s=1}^{14} x_{i_s j_t} = 1, \sum_{j_t=1}^{14} x_{i_s j_t} = 1, \sum_{i_s=1}^{14} x_{i_s, j_t, 1} = 1, \sum_{j_t=1}^{14} x_{i_s, j_t, 1} = 1, \\[2mm]
\displaystyle\sum_{i_s=1}^{14} x_{i_s, j_t, 2} = 1, \sum_{j_t=1}^{14} x_{i_s j_t, 2} = 1, \sum_{i_s=1}^{14} x_{i_s, j_t, 3} = 1, \sum_{j_t=1}^{14} x_{i_s j_t, 3} = 1, \\[2mm]
x_{i_s, j_t, 1} \neq x_{i_s, j_t, 2} \neq x_{i_s, j_t, 3}, x_{i_s j_t} \in \{0, 1\}, x_{i_s, j_t, 1} \in \{0, 1\}, x_{i_s, j_t, 2} \in \{0, 1\}, x_{i_s, j_t, 3} \in \{0, 1\}.
\end{cases}
\tag{40}
$$

The cost optimization model can be expressed as

$$
\begin{cases}
\gamma_3 = C_f \displaystyle\sum_{i_s=1}^{14} \sum_{j_t=1}^{14} x_{i_s j_t} \cdot l'_{i_s j_t} + \tau_\beta \cdot \left(C_{14}^3\right)^{-1} \cdot \displaystyle\sum_{i_s, j_t = 1}^{14} \left( x_{i_s, j_t, 1} \cdot x_{i_s, j_t, 2} \cdot x_{i_s, j_t, 3} \cdot r'_{\left(p_{i_1}, q_{j_1}\right), \left(p_{i_2}, q_{j_2}\right), \left(p_{i_3}, q_{j_3}\right)} \right), \\[2mm]
s.t. \displaystyle\sum_{i_s=1}^{14} x_{i_s j_t} = 1, \sum_{j_t=1}^{14} x_{i_s j_t} = 1, \sum_{i_s=1}^{14} x_{i_s, j_t, 1} = 1, \sum_{j_t=1}^{14} x_{i_s, j_t, 1} = 1, \\[2mm]
\displaystyle\sum_{i_s=1}^{14} x_{i_s, j_t, 2} = 1, \sum_{j_t=1}^{14} x_{i_s j_t, 2} = 1, \sum_{i_s=1}^{14} x_{i_s, j_t, 3} = 1, \sum_{j_t=1}^{14} x_{i_s j_t, 3} = 1, \\[2mm]
x_{i_s, j_t, 1} \neq x_{i_s, j_t, 2} \neq x_{i_s, j_t, 3}, x_{i_s j_t} \in \{0, 1\}, x_{i_s, j_t, 1} \in \{0, 1\}, x_{i_s, j_t, 2} \in \{0, 1\}, x_{i_s, j_t, 3} \in \{0, 1\}.
\end{cases}
\tag{41}
$$

## Optimization process using PNO–PGA

In this subsection, the introduced operation optimization problem would be solved by using the proposed novel PNO–PGA model. First of all, a series of parameters are obtained according to production practice. Specifically, in Eq. (21), it values that $\tau_v = 3.5$, $\tau_\alpha = 0.6$; in Eq. (41), it values that $C_f = 2.4$, $\tau_\beta = 0.3$. Then, by using the proposed PNO–PGA model, the problem is solved in the following manner.

**Step 1**: For any given feasible solution $x$ for Eq. (17), code $x$ and convert it into a vector $x'$ whose size is 14 and is a permutation from 1 to 14.

**Step 2**: Generate the initial population $P_0$ by using $x'$, and denote the population size of $P_0$ as $M$. Denote $X_0 = \{x_{0,1}, x_{0,2}, \ldots, x_{0,200}\}$, where $x_{0m}$ $(1 \leq m \leq 200)$ represents an individual. Denote $P_0 = \{x'_{0,1}, x'_{0,2}, \ldots, x'_{0,200}\}$, where $x'_{0m}$ $(1 \leq m \leq 200)$ is a vector which is corresponding to $x_{0m}$. Denote Group number = 200, cross rate = 0.8 and mutation rate = 0.2. The novel PNO–PGA will start the iteration with $P_0$. Set the iteration variable as $n$, and set the number of iterations as $N = 500$. Here $L'$ is a distance between 14 inbound and 14 outbound warehouses.

$$
L' = \begin{bmatrix}
0.13 & 0.18 & 0.13 & 0.24 & 0.28 & 0.28 & 0.32 & 0.34 & 0.54 & 0.51 & 0.57 & 0.54 & 0.77 & 0.77 \\
0.09 & 0.18 & 0.13 & 0.21 & 0.25 & 0.25 & 0.28 & 0.31 & 0.51 & 0.47 & 0.54 & 0.51 & 0.77 & 0.73 \\
0.09 & 0.18 & 0.13 & 0.18 & 0.21 & 0.21 & 0.25 & 0.28 & 0.47 & 0.43 & 0.51 & 0.47 & 0.77 & 0.73 \\
0.62 & 0.59 & 0.60 & 0.80 & 0.62 & 0.19 & 0.19 & 0.19 & 0.44 & 0.41 & 0.47 & 0.41 & 0.44 & 0.41 \\
0.09 & 0.18 & 0.13 & 0.18 & 0.21 & 0.21 & 0.25 & 0.28 & 0.47 & 0.43 & 0.51 & 0.47 & 0.77 & 0.73 \\
0.58 & 0.55 & 0.58 & 0.77 & 0.13 & 0.15 & 0.13 & 0.19 & 0.41 & 0.38 & 0.44 & 0.38 & 0.40 & 0.38 \\
0.55 & 0.52 & 0.55 & 0.74 & 0.09 & 0.12 & 0.13 & 0.19 & 0.38 & 0.34 & 0.41 & 0.34 & 0.37 & 0.34 \\
0.55 & 0.52 & 0.55 & 0.74 & 0.09 & 0.12 & 0.13 & 0.19 & 0.38 & 0.34 & 0.41 & 0.34 & 0.37 & 0.34 \\
0.44 & 0.41 & 0.41 & 0.44 & 0.06 & 0.06 & 0.09 & 0.03 & 0.19 & 0.19 & 0.19 & 0.22 & 0.23 & 0.20 \\
0.41 & 0.38 & 0.38 & 0.41 & 0.03 & 0.03 & 0.03 & 0.01 & 0.13 & 0.15 & 0.19 & 0.19 & 0.20 & 0.17 \\
0.31 & 0.28 & 0.28 & 0.31 & 0.01 & 0.01 & 0.03 & 0.01 & 0.09 & 0.12 & 0.19 & 0.15 & 0.17 & 0.13 \\
0.81 & 0.78 & 0.81 & 1 & 0.35 & 0.38 & 0.35 & 0.41 & 0.55 & 0.58 & 0.55 & 0.15 & 0.13 & 0.19 \\
0.75 & 0.72 & 0.75 & 0.93 & 0.29 & 0.32 & 0.29 & 0.34 & 0.49 & 0.52 & 0.49 & 0.09 & 0.06 & 0.12 \\
0.78 & 0.75 & 0.78 & 0.96 & 0.32 & 0.35 & 0.32 & 0.39 & 0.52 & 0.55 & 0.52 & 0.12 & 0.09 & 0.15
\end{bmatrix}.
$$

**Step 3**: For any given three individual $x_{0m}\,(1 \leq m \leq 200)$ in $P_0$, calculate its correspond-ing $E(x_{0m})$, $S(x_{0m})$, and $C(x_{0m})$ according to Eqs. (39), (40), and (41), respectively. Then, it gets a vector $V_{x_{0m}}$, a set of $G_{P_0}$, a set $V_{P_0}$, three parameters $E_{MAX,0}$, $S_{MAX,0}$, $C_{MAX,0}$. Thereafter, for any given $m\,(1 \leq m \leq 200)$, by using Eq. (23), it gets a vector $G_{x_{0m}}$ and a set $X_G$.

**Step 4**: For convenience, denote $\lambda = 2$. For any given $G_{x_{0m}}\,(1 \leq m \leq 200)$, by using Eq. (13), its corresponding GSF value can be obtained as $S_F\left(G_{x_{0m}}\right)$. Thereafter, by using $S_F(\cdot)$, a monotonic descending sequence is obtained as $l_{0F}$, where $m_1, m_2, \ldots, m_{200}$ isa permutation of $1, 2, \ldots, 200$. Here, set a positive integer $M^* = 160$. By using $M^*$, $X_0$ is divided into two parts, where one is $X_{01}$ with 160 elements, whereas the other is $X_{02}$ with 40 elements.

**Step 5**: By using certain crossover operator, $X_{01}$ is transferred to $X'_{01}$, where $X'_{01} = \left\{x'_{0m_1}, x'_{0m_2}, \ldots, x'_{0m_{160}}\right\}$,. By using mutation operator $X_{02}$ is transferred to $X'_{02}$, where $X'_{02} = \left\{x'_{0m_{M^*+1}}, x'_{0m_{M^*+2}}, \ldots, x'_{0m_{40}}\right\}$. Denote $X'_0 = \left\{X'_{01}, X'_{02}\right\}$ by using $S_F(\cdot)$, a monotonic descending sequence is obtained as $l'_{0F}$. For any given $m\,(1 \leq m \leq 200)$, it gets $S_F\left(G_{x_{1k}}\right)$ by using Eq. (30). Then, denote its corresponding set of feasible solution $X_1$, where $X_1 = \left\{x_{1,1}, x_{1,2}, \ldots, x_{1,200}\right\}$, anddenote the $1$st optimal solution as $x_{1,1}$.

**Step 6**: If $n \neq 500$, turn to Step 3. By calculating, it gets $E'_{MAX,1}$, $S'_{MAX,1}$, $C'_{MAX,1}$, $E_N\left(x_{1,m}\right)$, $S_N\left(x_{1,m}\right)$, $C_N\left(x_{1,m}\right)$. Then, turn to Step 4, Step 5, Step 6 in turn. If $n+1 < 500$, turn to Step 3; and denote $n = n+1$. Then, it gets $E'_{MAX,n}$, $S'_{MAX,n}$, $C'_{MAX,n}$, $E_N\left(x_{n,m}\right)$, $S_N\left(x_{n,m}\right)$ and $C_N\left(x_{n,m}\right)$. Then, turn to Step 4, Step 5, Step 6 in turn. If $n < 500$, turn to Step 3; if else, select the final optimal solution as $x'_{500,1}$.

The aforementioned program is experimented. At last, it gets the optimal vector as

$$
x'_{500,1} = \begin{bmatrix} 12 & 9 & 14 & 5 & 13 & 4 & 10 & 11 & 1 & 2 & 7 & 6 & 8 & 3 \end{bmatrix}.
$$

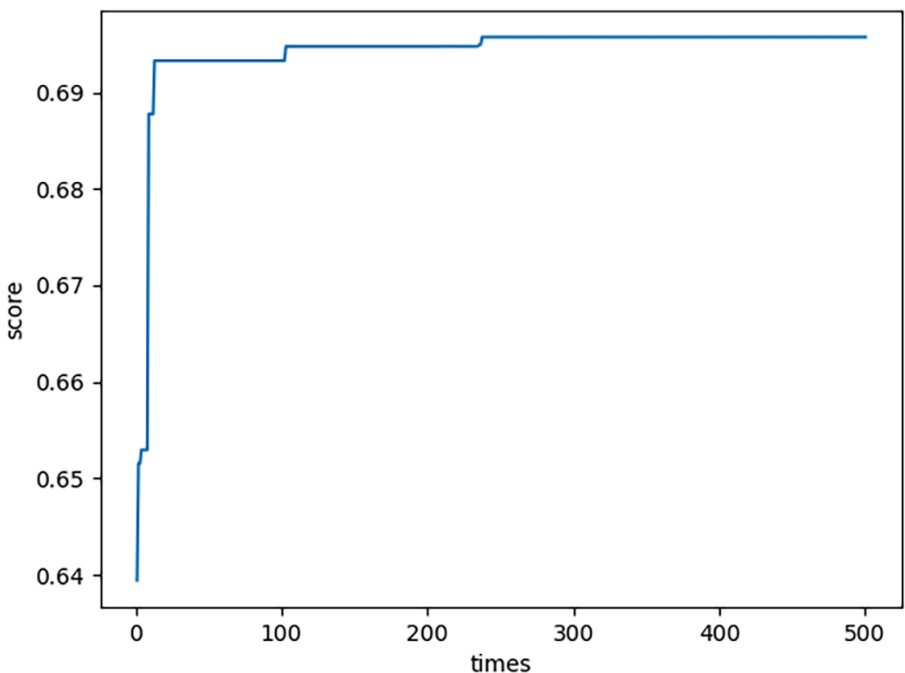

**Figure 4** Neutrosophic fitness values.

Then, the optimal feasible solution for the given question is obtained as

$$x_{500,1} = \begin{bmatrix} 1 & 2 & 3 & 4 & 5 & 6 & 7 & 8 & 9 & 10 & 11 & 12 & 13 & 14 \\ 12 & 9 & 14 & 5 & 13 & 4 & 10 & 11 & 1 & 2 & 7 & 6 & 8 & 3 \end{bmatrix}.$$

It is noteworthy that the serial number of inbound locations is represented by the first row of numbers in $x_{500,1}$, while the serial number of outgoing locations is represented by the second row of numbers in $x_{500,1}$.

## Description of calculation process

In this subsection, some peculiar characteristics in the calculation process in the previous subsection are described. (1) In each generation, there is a maximum neutrosophic fitness value $S_F(G_{x_{n,m1}})$ where $n = 1, 2, \ldots, 500$. It shows that $S_F(G_{x_{n,m1}})$ converges. More details please see Fig. 4. From Fig. 4, in the process of element and set coevolution, the feasible solution is optimized which illustrates the strategy of combining crossover and mutation effectively avoid the problems of premature convergence and local optimal solutions, thereby it improving the efficiency and accuracy of the optimization process. (2) In each generation, there is a feasible solution $x_{n,m1}$ which is corresponding to $S_F(G_{x_{n,m1}})$. By Eqs. (21)–(23), $E(x_{n,m1})$, $1 - S(x_{n,m1})$ and $C(x_{n,m1})$ are all obtained. The curve of $E(x_{n,m1})$ please see Fig. 5; the curve of $1 - S(x_{n,m1})$ please see Fig. 6; the curve of $C(x_{n,m1})$ please see Fig. 7. Specifically, Figs. 5, 6 and 7 show that $E(x_{n,m1})$, $S(x_{n,m1})$ and $C(x_{n,m1})$ all converge which illustrates the novel proposed PNO–PGA is effective. (3) In each generation, NS there are three maximum values in each domain, $i.e.$, $E_{MAX,n}$, $1 - S_{MAX,n}$, and $C_{MAX,n}$. By using functions similar to Eq. (22), they are all obtained. The

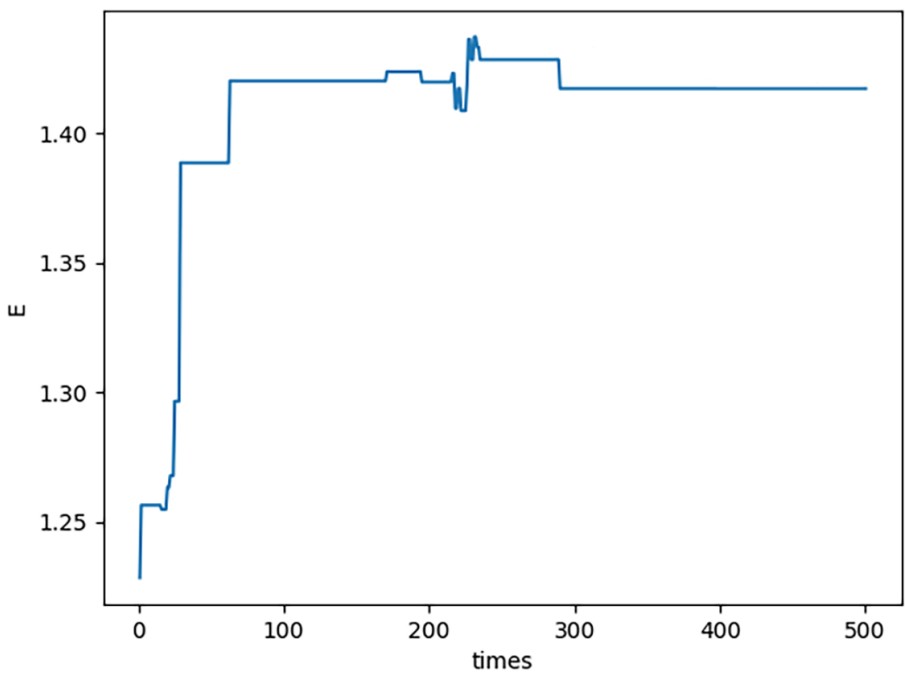

**Figure 5** E(Xn,m1)for each generation.

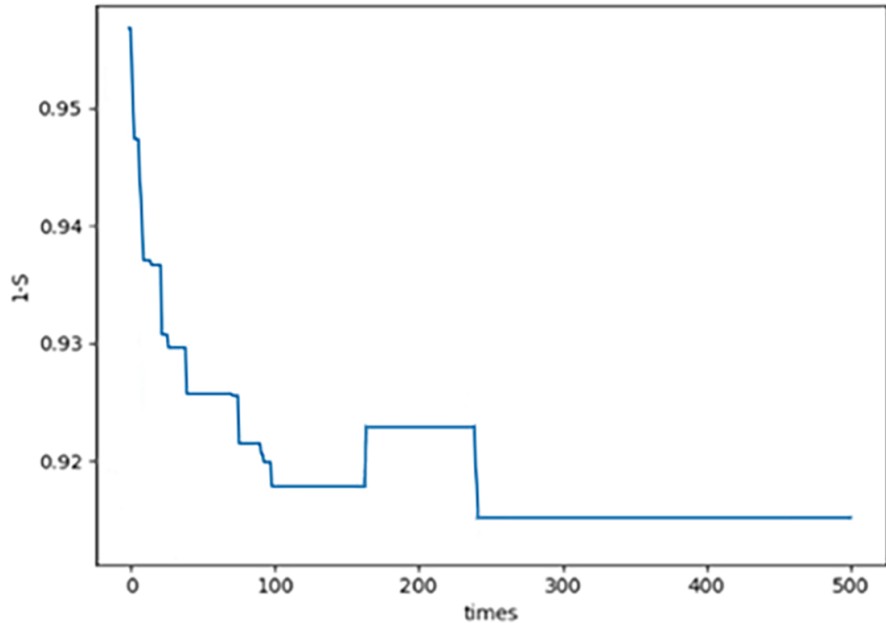

**Figure 6** 1-S(Xn,m2) for each generation.

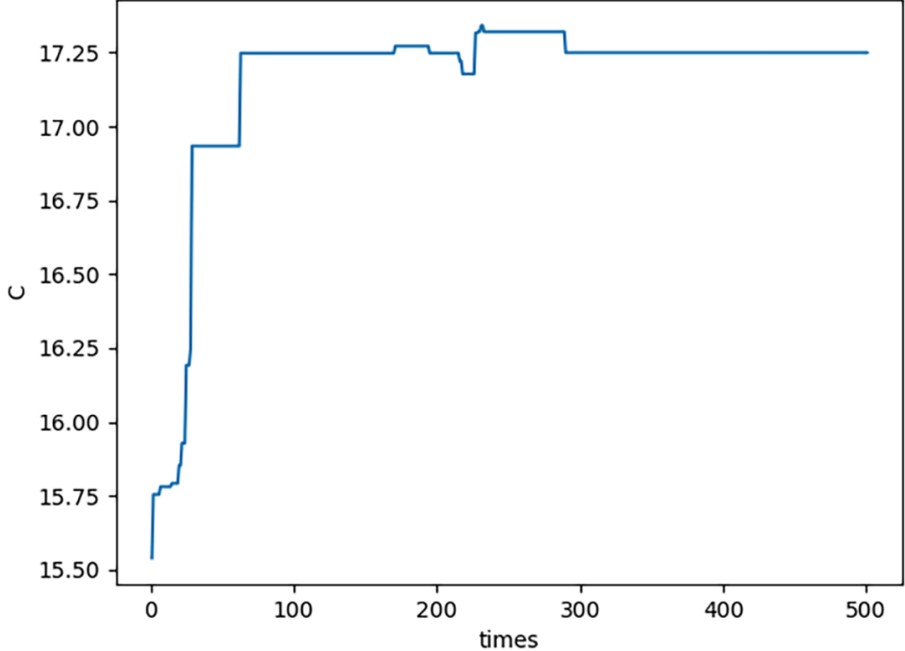

**Figure 7** C(Xn,m3) for each generation.

curve of $E_{MAX,n}$ please see Fig. 8; the curve of $(1 - S)_{MAX,n}$ please see Fig. 9; the curve of $C_{MAX,n}$ please see Fig. 10. Specifically, Figs. 8, 9 and 10 show that $E_{MAX,n}$, $(1 - S)_{MAX,n}$, and $C_{MAX,n}$ all converge which illustrates the definition of s is effective. (4) In the proposed PNO–PGA, there are two key parameters, *i.e.,* λ and $M^*$. On the one hand, when the customer cares more about emergency situation, the parameter λ should be valued as a large number. To the contrary, when the customer cares more about normal production, the parameter λ should be valued as a small number. On the other hand, when the customer thinks the complexity of the research problem is strong, especially when the nonlinearity of the model is strong, the parameter $M^*$ should be valued as a small number. To the contrary, when the customer thinks the complexity of the research problem is weak, especially when the linearity of the model is strong, the parameter $M^*$ should be valued as a large number.

(5) The introduced MOO problem can also be solved by using classical GA. However, comparing with classical GA, the proposed PNO–PGA has stronger search and anti-precocity ability. Specifically, the crossover and variation in PNO–PGA are executed by in proportion rather than probability, so that the population generally evolved over and over again. The comparison between the quality of the first-generation individuals and that of the last-generation individuals in the mutation area is shown in Fig. 11. The performance of the classical GA and the PNO-PGA in Mean convergence iterations and Standard deviation of CPU time are compared and analyzed by applying the benchmark test case ten times with 500 iterations per iteration, respectively. The experimental results show that the Mean convergence iterations of classical GA was 237.55 times, and PNO-PGA

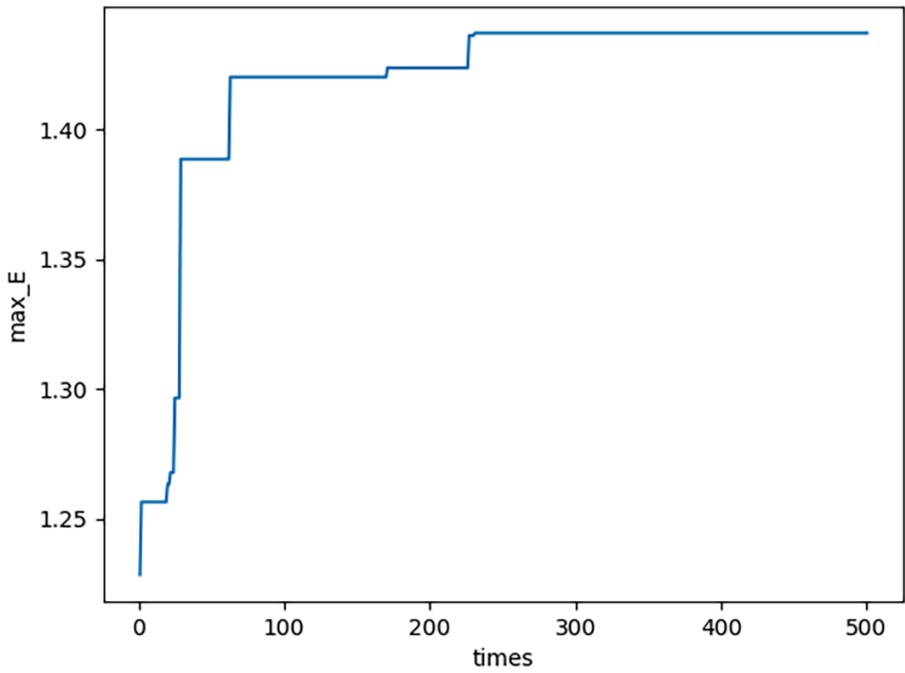

**Figure 8** E max,n for each generation.

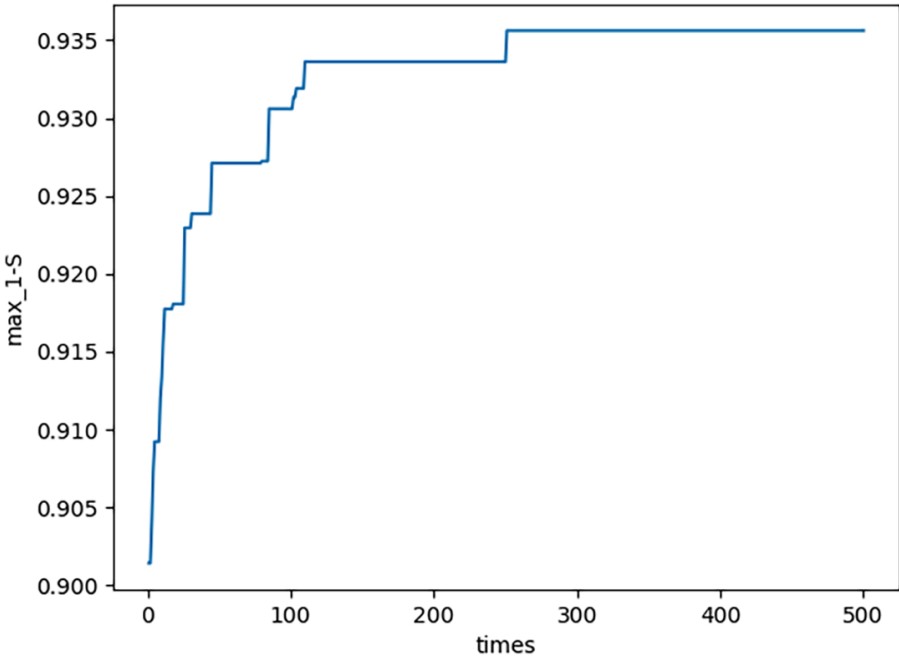

**Figure 9** (1-S)max,n for each generation.

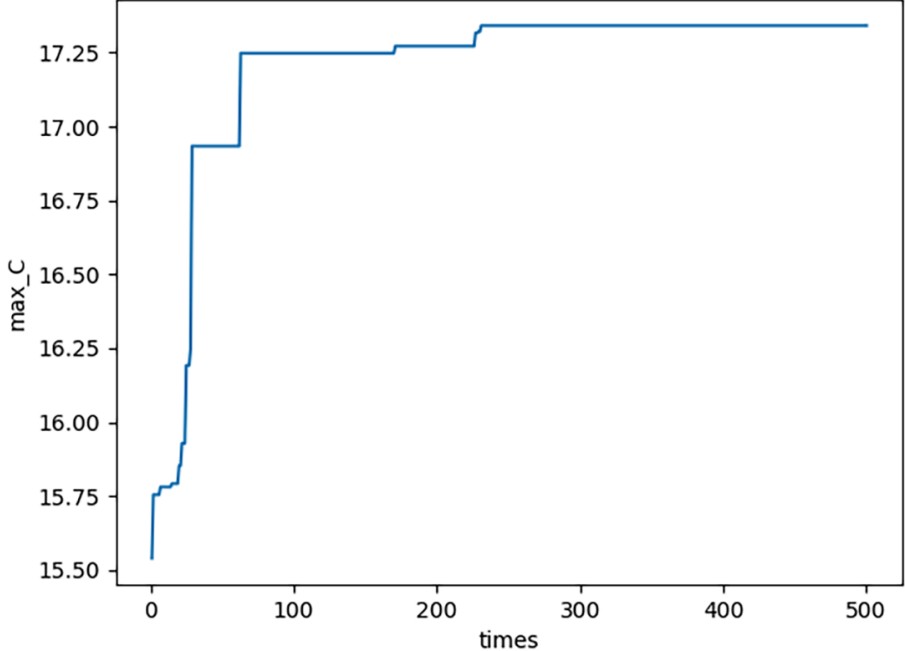

**Figure 10  Cmax,n for each generation.**

exceeded GA 152 times, which demonstrates that PNO-PGA is more stable in searching ability and more accurate in finding the global optimal solution. In the Standard deviation of CPU time, PNO-PGA's 6.39 s is shorter than that of classical GA's 25.05 s by 18.66 s. it is noteworthy that PNO-PGA is more excellent than classical GA in terms of search efficiency. The reliability and utility of the algorithm's performance enable it not to be affected by input data or other factors. The experimental comparison results of PNO-PGA and GA are shown in Table 1.

## CONCLUSION

In this study, a kind of neutrosophic objective optimization thought is proposed whose characteristic is to dialectically monitor the whole process of optimization activities by using neutrosophic thought. According to this thought, a novel parameterized PNO-PGA is proposed. Finally, through an example, the validity of the proposed PNO-PGA is verified, and the proposed PNO-PGA has three main characteristics.

Firstly, compared with traditional GA, the proposed PNO-PGA cleverly utilizes NSs to deal with the three main objectives of industrial production, *i.e.,* efficiency, safety, and cost. The advantage is that it is more adaptable to more fuzzy and generalized situations.

Secondly, the improved GA can explore the potential solution space better through proper crossover and variation proportion, maintain population diversity, and avoid the search process from falling into premature, locally convergent scenarios.

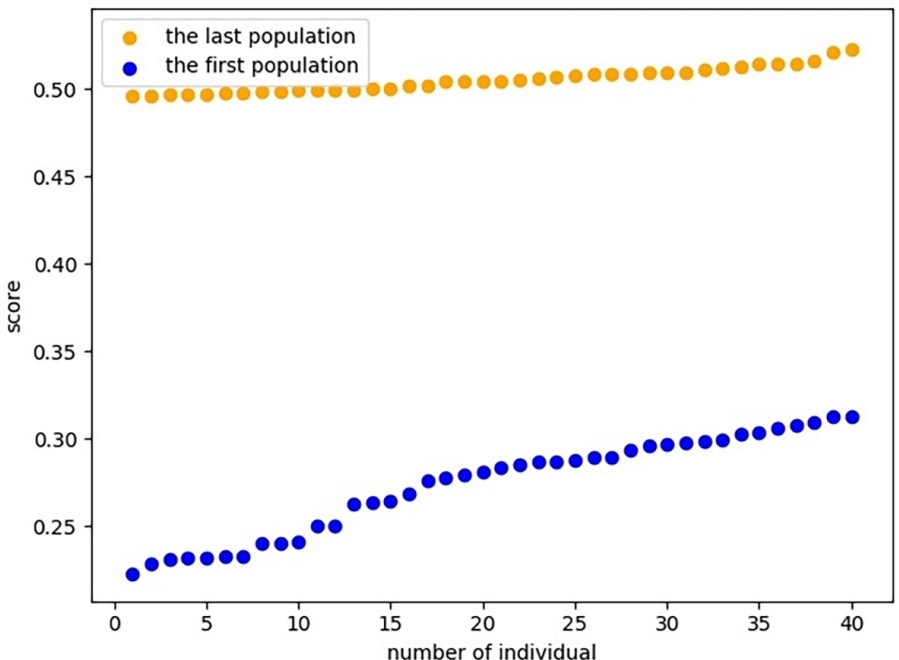

**Figure 11  Evolutionary effects of individuals in variation area.**

Thirdly, based on the characteristics of the problem and the progress of the algorithm, parameterized GA can flexibly modify the weights of the objective function, dynamically. In addition, To integrate subjective and objective information by using the two parameters effectively.

The proposed PNO–PGA is an important cornerstone. By using this algorithm, NS theory and optimization theory have been held together tightly for the first time. In the future, similar to PNO–PGA, it can deduce a series of algorithms for MOO such as neutrosophic objective-neural networks, neutrosophic objective-particle swarm optimization, neutrosophic objective-ant colony algorithm, *etc.*

The model expands the range of objective value options by aggregating vectors of objectives and changing the weights. The model can integrate objective values, dialectically monitor the whole process of optimization activities, and have potential applications in business management, vehicle routing problems, and so on. For the different optimization objectives of decision makers in different scenarios, the specific implementation means in terms of weight selection are different, and the actual optimization situation is obtained through practical investigations. In the future, further optimization of GA is planned to expand its applicability in the whole process of optimization. Moreover, the model can be optimized more by using other aggregation operators and can be more realistic in other industries and engineering fields.

### Funding

This work was supported by the Shanghai Puiang Program (No. 2019PJC062), the Research project on Undergraduate Teaching Reform of higher education in Shandong Province (No. Z2021046), the Natural Science Foundation of Shandong Province (No. ZR2021MG003), the National Natural Science Foundation of China (No. 51508319), and the Nature and Science Fund from Zhejiang Province Ministry of Education (No. Y201327642). Natural Foundation of Shandong Province (ZR2021MG003) provided assistance in data collection and analysis.

### Grant Disclosures

The following grant information was disclosed by the authors:
The Shanghai Puiang Program: 2019PJC062.
The Research project on Undergraduate Teaching Reform of higher education in Shandong Province: Z2021046.
The Natural Science Foundation of Shandong Province: ZR2021MG003.
The National Natural Science Foundation of China: 51508319.
The Nature and Science Fund from Zhejiang Province Ministry of Education: Y201327642.
Natural Foundation of Shandong Province: ZR2021MG003.

### Competing Interests

Jun Ye is an Academic Editor for PeerJ.

### Author Contributions

- Yi Zhao conceived and designed the experiments, performed the experiments, analyzed the data, performed the computation work, authored or reviewed drafts of the article, and approved the final draft.
- Fangwei Zhang performed the experiments, analyzed the data, authored or reviewed drafts of the article, and approved the final draft.
- Bing Han analyzed the data, performed the computation work, prepared figures and/or tables, and approved the final draft.
- Jun Ye performed the computation work, prepared figures and/or tables, and approved the final draft.
- Jingyuan Li conceived and designed the experiments, performed the computation work, prepared figures and/or tables, and approved the final draft.

### Data Availability

 The raw measurements are available in the Supplementary File.

## Supplemental Information

Supplemental information for this article can be found online at http://dx.doi.org/10.7717/peerj-cs.2117#supplemental-information.

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
