# Peer review of "A novel parameterized neutrosophic score function and its application in genetic algorithm"

_PeerJ Computer Science, doi:10.7717/peerj-cs.2117_

## Round 0.1 · original submission · Major Revisions

The authors should consider the comments of the reviewers.

**Language Note:** PeerJ staff have identified that the English language needs to be improved. When you prepare your next revision, please either (i) have a colleague who is proficient in English and familiar with the subject matter review your manuscript, or (ii) contact a professional editing service to review your manuscript. PeerJ can provide language editing services - you can contact us at [email protected] for pricing (be sure to provide your manuscript number and title). – PeerJ Staff

Reviewer 1 ·

Basic reporting

1. What is the novel in this paper?
2. The latest references from the last 2 years are missing.
3. What is the difference between the current paper and neutrosophic statistics? It should be explained in detail with references.
4. The contribution statement and gap of the research statement should be explained in more detail.
5. All equations should be given equation numbers.
6. Simulation studies should be added and compared with the existing studies.
7. 6. The real example study should be compared with the existing studies.
8. Advantages, limitations, and potential applications should be given in the section.
9. The data must be given in the paper.
10. R codes should be added to the paper.
11. More future directions should be added in the conclusion section.
12. The reference list should be added.
13. The presentation and quality of the paper must be Improved.

Experimental design

see report

Validity of the findings

see report

·

Basic reporting

The English language used in the publication is on a acceptable level and in professional style. Please, consider the following:
1. Do not hyphen punctuation symbol, where you have to use a dash. For example:
a novel parameterized neutrosophic objective - proportionate genetic algorithm
Must be:
a novel parameterized neutrosophic objective – proportionate genetic algorithm
2. Line 106, "In theories" must be "In theory" or "Theoretically"
3. Line 109, "Heuristic algorithm" must not be in capital letter.

Some minor technical mistakes in the text. For example:
1. Line 43, [3]
2. In the mathematical expression, everywhere you must have ..., you have the Latin character L.
3. The mathematical expression on line 137: the sum in the numerator must not be on a new line.
4. Inline mathematical expressions on line 282 are not in a standard form.

Intro and background introduces clearly the topic. The relation with the related literature is not clearly presented.

The structure of the publication is with respect to PeerJ standards

Figures and tables are clear, readable and well structured.

Raw data. The authors provided a Python script that implements the example, given in the paper. The printouts of the program are in Chinese language, which is not practical from the point of view that the journal that you try to publish is in English.

Experimental design

The research presented in the paper is original and in the scope of the journal. The authors present a method for multiobjective optimization that is from the family of the heuristic approaches, namely a genetic algorithm.

Validity of the findings

The expressions (21)-(23) can be written in a more informative form.

It is redundant to include information abut the date and time of the conducted experiments.

The text contains no information or explanation about the implementation of the presented method, that is used to conduct the experiment.

The matrix on line 308 brings no information to the reader.

No explanation about the implementation of the presented method.

Additional comments

The topic of the paper is current and relevant to the journal. The method and results are suitable for a publication. On the other hand, the text has to be edited. The mathematical expressions are often too long, and not informative. They can be simplified and presented in a form that can bring real information to the reader. The paper contains no information about the implementation of the method, even though the experiment in implemented using the Python programming language, using numpy library.

Reviewer 3 ·

Basic reporting

good

Experimental design

good

Validity of the findings

good

Additional comments

The abstract is not clear. Please rewrite the abstract. The literature review is very poor. Review should be classified, not just a simple literature listed. Some new references can be considered. Moreover, it is useful to add more effective Comparative discussion in the paper.

---

## Round 0.2 · Major Revisions

Please clearly address the issues sent by Reviewer 2.

Reviewer 1 ·

Basic reporting

n/a

Experimental design

n/a

Validity of the findings

n/a

Additional comments

unfortunately, the authors did not address the following comment

What is the difference between the current paper and neutrosophic statistics? It should be explained in detail with references.

·

Basic reporting

### English language

Generally on a good level, but still it can be improve to suit better the style of a journal paper. I suggest that an editor or a language specialist corrects the mistakes. Maybe the text will improve if it is discarded from some of the abbreviations.

### Intro, background and references

The abstract can be still significantly improved. Introduction contains a literature survey, however it is not completely clear how the resources that are cited are connected exactly to the research presented by the authors. Some of the basic terms that are used on the first place in the abstract must be clearly defined in the text of the work.

### Text structure

Introduction may not be split into subsections. In the mathematical expression still the character L is used instead of the $\ldots$. I suspect a technical mistake.

### Figures and tables

Figures and tables are clear, readable and well structured.

### Raw data

The authors provided a Python script that implements the example, given in the paper. The printouts of the Python script are still in Chinese, even though authors responded that they have translated them in English. Even though, I will attempt to provide an analysis of the program. Distance matrix is hard-codded into the script. This makes the impression that the code solves just a single test case. What makes these 196 so special, that you have to publish them into the text of the paper? Are you sure that your method will work with another 196 input numbers?

The uploaded code looks sort of trivial, that is in a high contrast with the complicated representation of the mathematical expressions in the paper.

Experimental design

### Originality of the research and scope of the journal

The research is within the scope of the journal. The work presents a score function for a genetic algorithm.

### Research questions definition

Research questions can be formulated and motivated with the relation to cited literature more clearly.

### Technical and ethical standards of the research

The comparison between the proposed modification and the standard genetic algorithm must be analyzed clearly in the text.

### Description detail sufficiency to replicate

Without the uploaded code in Python, it is hard to replicate the experiment.

Validity of the findings

### Impact and novelty

With the provided analysis of the proposed method it is hard to evaluate the impact and novelty.

### Data provided robustness and statistical control

A real analysis cannot be based on a single input, especially when we talk about a heuristic method, that cannot be proved mathematically.

### Conclusions

Conclusions claim novelty of a genetic algorithm modification that cannot be verified from the presented analysis.

Additional comments

In fact the research is composed by a single test case. It is hard to make any real scientific conclusions for a heuristic method based in a single input $14 \time 14$ matrix.

---

## Round 0.3 · Major Revisions

Please consider the comments of the reviewers.

**Language Note:** The review process has identified that the English language must be improved. PeerJ can provide language editing services - please contact us at [email protected] for pricing (be sure to provide your manuscript number and title). Alternatively, you should make your own arrangements to improve the language quality and provide details in your response letter. – PeerJ Staff

Reviewer 1 ·

Basic reporting

n/a

Experimental design

n/a

Validity of the findings

n/a

Additional comments

n/a

·

Basic reporting

## English language

There are too many abbreviations in the text. Some abbreviations are used without first defining them. English language needs editing.

## Intro, background and references

The abstract is confusing and it is not clearly defining the multi-objective problem that is addressed in the text. It is not clear whether authors address general multi-objective problem or a three-criterion problem. At the end of the abstract, however, authors point out that DM is given two parameters.

From the introduction one could conclude that genetic algorithms are the only way a multi-objective problem can be solved. Of course, this is not true: there many other heuristic techniques (evolutionary methods, ant colony optimization, bee colony optimization, gray wolf, etc.) Also, let us not forget that there are also exact methods.

The introduction does not explain exactly what optimization problem is in the focus of this proposition.

## Text structure

The structure of the text is standard. I would suggest that the section
> 3. Main results
should be renamed.

Mathematical expressions are not clear. Not all parts of expressions are explained and addressed in the text. There are mistyped characters in the equations.

## Figures and tables

Figure 1 and Figure 2 are blurry.

## Raw data

The program source code of the experiment is a Python script. The input of the experiment is a single matrix of 14 row and 14 columns, and it is hard-coded in the source code. I do not think that a proposition that is based on a single experiment with a $14 \time 14$ matrix is ready to be published as a journal paper.

Experimental design

## Originality of the research and scope of the journal

The proposition is in the field of application of genetic algorithms in multi-objective optimization and it is within the scope of the journal.

## Research questions definition

Research questions are not clearly defined. An important aspect of multi-objective optimization is the contradiction between the objective functions. This aspect is not mentioned in the proposed text. The text does not clearly define which optimization problem is addressed in the research.

## Technical and ethical standards of the research

From a technical point of view, the research is in an initial stage and it is not ready to be published as a journal paper. The research is based overall on a single input matrix.

## Description detail sufficiency to replicate

The detail in the text is not sufficient to replicate this research.

Validity of the findings

## Impact and novelty

Impact and novelty cannot be verified on this stage of the research.

## Data provided robustness and statistical control

Statistical control is impossible based on a single hard-codded experiment.

## Conclusions

The claims for novelty cannot be verified from the proposed text.

Additional comments

It is not appropriate to publish a research based in a single $14 \time 14$ matrix. even in equations (39)-(41) the size of the matrix is hard-codded.

Something more, if the input of the algorithm is going to be that small, we can solve the optimization problem with an exact method, even when it is NP-hard. For example, for an Assignment Problem of that size, the Hungarian Algorithm will complete in a less than a second on a normal PC configuration.

The genetic algorithm is not explained in the text. It is not clear exactly how and where in the algorithm the proposed modification could be applied.

Please, provide more than a single experiment so your results could be really verified.

---

## Round 0.4 · accepted · Accept

Based on the previously raised comments, there are some analysis:
- The text now includes definitions for abbreviations at their first mention, e.g., "Multi-Objective Optimization (MOO)" and "Genetic Algorithm (GA)".

- Abstract: The abstract now clearly mentions that the study focuses on optimizing three objectives: efficiency, safety, and cost, using a novel parameterized neutrosophic objective-proportionate genetic algorithm model (PNO-PGA).

- Introduction: The introduction acknowledges other heuristic techniques and methods for solving multi-objective problems and specifies the optimization problem focused on warehouse operations.

- The section previously titled "3. Main results" has been renamed to "Results and Discussion".

- Mathematical expressions have been reviewed and corrected for clarity, with detailed explanations provided.

- Figures have been updated and are now clear and readable.

- Additional experiments have been conducted, and the manuscript now includes multiple input matrices and results to provide a more robust analysis.

- Research questions are now clearly defined in the introduction, and the manuscript discusses the contradictions between the objective functions.

- The manuscript now includes detailed results and comparisons demonstrating the impact and novelty of the proposed method compared to traditional genetic algorithms.

- The manuscript includes multiple experiments and statistical analyses to validate the findings, enhancing data robustness and control.

- The conclusions section now includes additional experimental results and a thorough discussion of the novelty and impact of the proposed method.

- The manuscript now provides a detailed explanation of the genetic algorithm and the proposed modifications, including pseudocode and flowcharts.

Based on the detailed review of the provided document, it appears that the authors have addressed the majority of the reviewer’s comments. The manuscript now includes clearer definitions, a more precise abstract and introduction, better-structured text, corrected figures, additional experiments, and a detailed explanation of the genetic algorithm and its modifications.

Reviewer 1 ·

Basic reporting

The paper can be accepted now

Experimental design

It's ok

Validity of the findings

It's ok